# Surface Hydrophobization of Block-Shaped Wood with Rapid Benzylation

Mitsuru Abe * , Masako Seki , Tsunehisa Miki  and Masakazu Nishida

Multi-Materials Research Institute, National Institute of Advanced Industrial Science and Technology (AIST), 2266-98 Shimoshidami, Moriyamaku, Nagoya 463-8560, Japan; m-seki@aist.go.jp (M.S.); tsune-miki@aist.go.jp (T.M.); m-nishida@aist.go.jp (M.N.)
* Correspondence: m.abe@aist.go.jp; Tel.: +81-52-736-7209

**Abstract:** With the aim of utilizing wood as a carbon cycle-oriented material, the improvement of hydrophobicity has been actively studied to solve manufacturing problems, such as dimensional stability and biodeterioration resistance. The introduction of benzyl group is a promising chemical modification for hydrophobizing wood. However, conventional benzylation methods are not suitable for industrial applications because they require high temperature and long reaction times. In this study, a novel method was developed for quickly benzylating the surface of block-shaped wood using an aqueous solution of tetra-*n*-butylphosphonium hydroxide as a pretreatment solvent and no heat. The color and shape of the benzylated wood was almost unchanged from that before the treatment. Analysis of the resulting chemical structure suggested that the developed method causes less damage to carbohydrates compared with the conventional method, which involves heating and stirring. The proposed method successfully imparted hydrophobicity and thermoplasticity to the benzylated wood surface. Furthermore, hydrophobicity of the benzylated wood was further improved by a simple heat treatment for only approximately 5 min. The water contact angle became $\geq 110°$ and remained almost unchanged even after 1 min after water dropping.

**Keywords:** wood surface; tetra-*n*-butylphosphonium hydroxide; less degradation; attenuated total reflection infrared; solid-state nuclear magnetic resonance; dynamic mechanical analysis; contact angle; scanning electron microscopy

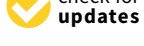



## 1. Introduction

Wood is a renewable carbon cycle material and it requires further effective utilization for the construction of a sustainable society. However, wood has various problems such as low dimensional stability, perishability, and flammability, which limit the industrial use of wood. To overcome these problems, various modifications have been studied and some results have been achieved [1–3]. However, many issues remain, such as the complexity of processing and the environment impact of the process.

In particular, the water resistance of wood is one of the major problems in its use as an industrial product because it is closely related to the dimensional stability and biodeterioration resistance of wood. These issues can be addressed by hydrophobizing the wood. Various methods for imparting water resistance to wood with environmentally friendly way have been studied, e.g., copper-resin impregnation [4] and ZnO-coating [5]. Among these, chemical modification is a promising way to hydrophobize woods. Compared to common methods such as impregnating other ingredients in wood, chemical modification has the advantage that components introduced into the wood, often substances that are harmful to the environment, are less likely to seep into the environment [6]. Acetylation treatment of wood is a popular chemical modification for converting hydrophilic hydroxyl groups to hydrophobic acetyl groups. Acetylation is a relatively simple chemical treatment investigated in many studies so far. Acetylated wood has been put to practical use in various fields [7]. However, when focusing on increasing hydrophobicity, the use of long-chain

esters has shown better results than the use of acetyl groups. Thiebaud et al. esterified oakwood using fatty acids with different chain lengths and their chlorides [8]. These reactions were carried out at high temperatures in a nitrogen atmosphere. The weight gain reached 6% for the highest grafting yields, and the water contact angle of the obtained wood was approximately 95°. Bach et al. succeeded in imparting even higher hydrophobicity to wood by combining esterification with further chemical treatment. They used esterification and copolymerization using styrene (ST) and methyl methacrylate (MMA) to hydrophobize wood [9]. A pine wood sample was esterified with pyromellitic anhydride in a dry non-swelling medium, and then, it was copolymerized with ST or MMA. The water contact angle of the chemically modified wood treated with MMA was approximately 100°, whereas that of the original untreated wood surface was approximately 40° just after the drop deposition test, and it decreased to zero within a few seconds. In contrast, ether groups are generally more stable than ester groups, that is, the hydrophobicity of wood has been investigated by not only esterification but also etherification followed by copolymerization [9]. After an etherification treatment that reacts pine wood with glycidyl methacrylate, these modified woods were copolymerized with ST or MMA to obtain hydrophobic resin copolymerized woods. The water contact angles on these modified wood surfaces were >110°. Several papers have reported successful examples of imparting hydrophobicity to wood using chemical modifications. However, these modification methods have not been used industrially because the processing is complicated and takes a long time. Therefore, it is necessary to develop easier and faster ways for imparting hydrophobicity to wood.

When considering the chemical modification of wood, the depth of chemical treatment is another important issue. Uniform chemical modification of the interior of wood is challenging and generally a complex and time-consuming process. Moreover, chemical modification of wood can alter the properties of the original wood, for better or for worse, sometimes leading to new problems in the modified wood [7,10]. For example, if the humidity control ability of the wood is reduced, dew condensation and mold are generated when these modified woods are used as interior materials, which creates an unfavorable situation in terms of health and hygiene. Therefore, as an attempt to take advantage of the characteristics of both natural wood and chemically modified wood, surface etherification of wood is being studied. Benzylation is a promising chemical modification method that can impart thermoplasticity, water resistance, dimensional stability, and biodeterioration resistance to wood [11]. Among various chemical modifications, the acetylated wood has been studies most actively and is being put into practical uses. Compared to the acetylated wood, the benzylated wood has advantages such as high hydrophobicity and ultraviolet absorption capacity. However, the benzylation is less reactive than the acetylation and requires a longer stirring at higher temperature. In the conventional method, the surface of wood is generally pretreated with a high concentration sodium hydroxide (NaOH) aqueous solution, followed by immersion in a benzylating reagent with stirring at approximately 120 °C for several hours. Because it requires long heat treatment in a closed system at the laboratory level, there are restrictions on the size of wood that can be treated. In addition, depending on the pretreatment conditions and the subsequent benzylation treatment, several problems may occur, such as undesired blackening or whitening of the wood and roughening of the wood surface [12]. For these reasons, it is difficult to use the conventional benzylation methods for the industrial process.

Recently, we developed a novel process for rapid benzylation of wood flour without heating [13]. In that study, a translucent film was obtained by heat pressing. This technique can be applied to block-shaped wood to impart hydrophobicity by simple treatment. Notably, the treatment is expected to proceed in a short time without heating. This method has the potential to benzylize wood in a very simple process of coating and washing only. Furthermore, it is possible to easily apply chemical modifications that impart hydrophobicity to large-area wood materials and already-molded wood products, which is difficult with conventional methods. Aiming for wood production and promotion using chemical

modification, this study develops a method for efficiently imparting hydrophobicity by benzylating block-shaped wood using a simple treatment. We investigated the effects of the species of chemical reagents, treatment temperature, and time. Thereafter, the chemical structure of the treated wood was analyzed using attenuated total reflection infrared (ATR-IR) spectroscopy and solid-state nuclear magnetic resonance (NMR) spectroscopy. Furthermore, we report that we successfully obtained benzylated wood showing excellent water repellency by short-time heat treatment based on the thermoplasticity of the benzylated surface.

## 2. Materials and Methods

### 2.1. Original Materials

As a block-shaped wood, Japanese cedar (*Cryptomeria japonica*) was used because it is the most common coniferous wood as a structural material in Japan among various types of wood. The original Japanese cedar timber was logged in Kyoto prefecture and was purchased from Science Shokai, LLC. (Nagoya, Japan). The dimensions of the wood specimen were 5.0 mm × 20 mm × 20 mm, radial direction (R) × longitudinal direction (L) × tangential direction (T), respectively, and initial mass was ~0.7 g. The specimens were cut from a larger piece with a transverse section of 20 mm (L) × 20 mm (T). Methanol, NaOH, benzyl chloride (BnCl), and benzyl bromide (BnBr) were purchased from FUJIFILM Wako Pure Chemical Corp. (Osaka, Japan) and used as received. A 40% aqueous tetra-*n*-butylphosphonium hydroxide ($[(n\text{-}Bu)_4P]OH$) solution was purchased from the same company and concentrated to 50% at 35 °C under vacuum before use.

### 2.2. Preparation of Wood Samples

To degrease the wood specimens, Soxhlet extraction was carried out using methanol for 24 h and then hot water for 24 h. Then, the sample was washed with distilled water. The degreased wood was dried at 35 °C under vacuum for 24 h (entry 0 in Table 1).

**Table 1.** Treatment conditions of benzylation and the intensity ratio of the ATR-IR peaks.

| Sample | Alkaline Treatment | | | Benzylation Treatment | | | Intensity Ratio of IR Peaks [a] | |
|---|---|---|---|---|---|---|---|---|
| | Solvent | Temp (°C) | Time (min) | Reagent | Temp (°C) | Time (min) | OH/CH | Bn/CH |
| entry 0 [b] | - | - | - | - | - | - | $3.6 \pm 0.14$ | $0.3 \pm 0.08$ |
| entry 1 | 40% NaOH | 25 | 60 | BnCl | 110 | 120 | $0.7 \pm 0.06$ | $9.4 \pm 0.63$ |
| entry 2 | 40% NaOH | 25 | 60 | BnCl | 25 | 10 | $3.1 \pm 0.04$ | $0.3 \pm 0.06$ |
| entry 3 | 40% NaOH | 25 | 60 | BnBr | 25 | 10 | $3.3 \pm 0.08$ | $0.5 \pm 0.26$ |
| entry 4 | 50% ($(n\text{-}Bu)_4P$)OH | 25 | 10 | BnCl | 25 | 10 | $1.7 \pm 0.13$ | $3.1 \pm 0.25$ |
| entry 5 | 50% ($(n\text{-}Bu)_4P$)OH | 25 | 10 | BnCl | 25 | 30 | $1.4 \pm 0.05$ | $7.9 \pm 1.24$ |
| entry 6 | 50% ($(n\text{-}Bu)_4P$)OH | 25 | 10 | BnBr | 25 | 10 | $0.6 \pm 0.08$ | $8.1 \pm 0.36$ |
| entry 7 | 50% ($(n\text{-}Bu)_4P$)OH | 25 | 10 | BnBr | 25 | 30 | $0.6 \pm 0.02$ | $8.7 \pm 0.27$ |
| entry 8 | 50% ($(n\text{-}Bu)_4P$)OH | 25 | 10 | BnBr | 25 | 60 | $0.6 \pm 0.04$ | $9.3 \pm 0.23$ |
| entry 9 | 50% ($(n\text{-}Bu)_4P$)OH | 25 | 60 | BnBr | 25 | 60 | $0.5 \pm 0.03$ | $8.3 \pm 0.32$ |

[a] The standard deviations were calculated based on the values of three samples.; [b] Only degreasing treatment was conducted for the wood sample.

The typical procedure for the benzylation method is as follows. A degreased wood piece was placed on 0.5 mL of a 50% aqueous $[(n\text{-}Bu)_4P]OH$ solution at 25 °C. Because the amount of the solution was very small, the solution was only present under and around the piece of the wood as illustrated in Figure 1. Subsequently, the wood specimen was placed on 0.5 mL of a benzylation reagent at 25 °C. The benzylated wood specimen was washed by Soxhlet extraction using methanol for 24 h and then hot water for 24 h. The obtained wood was dried at 35 °C under vacuum for 24 h (Figure 1).

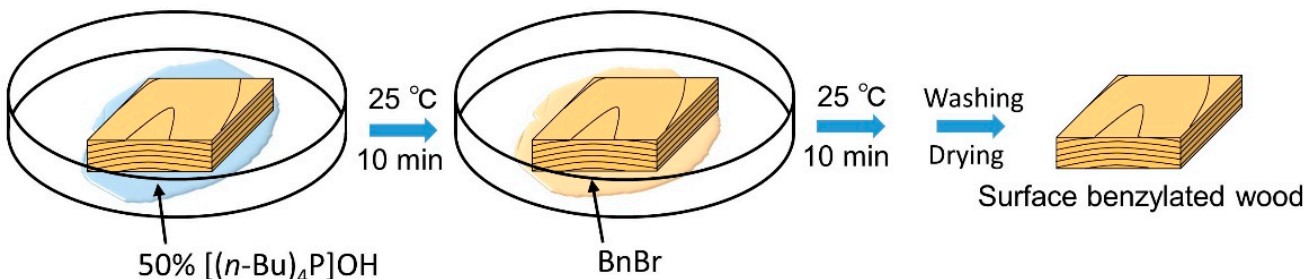

**Figure 1.** A typical benzylation process of block-shaped wood in this study.

For comparison, another benzylated wood sample was produced using a conventional method with heating. With reference to the report by Kiguchi et al. [14], the experiment was carried out by partially reducing the amount of the reagent. A piece of wood was placed on 0.5 mL of a 40% NaOH solution at 25 °C for 1 h. Then, the piece of wood was put into a separable flask. Thereafter, 4 mL of BnCl was added to the flask, and it was heated to 110 °C for 2 h. The washing and drying procedure is the same as the method using [(*n*-Bu)₄P]OH solution.

### 2.3. ATR-IR Spectroscopy

ATR-IR measurements were performed on a Nicolet 6700 spectrometer (Thermo Scientific Inc., Waltham, MA, USA) with a resolution of 4 cm$^{-1}$ in the standard attenuated total reflectance mode. In total, 32 scans were made in the range of 4000–500 cm$^{-1}$. The measurements were carried out at three or more points on the test piece, and no noticeable differences in the results were observed.

### 2.4. Solid State NMR Spectroscopy

Magic-angle spinning (MAS) NMR spectra were measured on a Varian 400 NMR system spectrometer (Varian Inc., Palo Alto, CA, USA) with a Varian 4 mm double-resonance T3 solid probe. A thin sample (approximately 0.1 mm) was cut out from near the surface, shredded, and then used for NMR measurement. The samples were placed in a 4 mm ZrO$_2$ rotor spun at 15 kHz within a temperature range of 24–26 °C. $^{1}$H MAS NMR spectra were collected with a 2.9 μs π/2 pulse at 399.87 MHz for the $^{1}$H nuclei and were collected with a 40 ms acquisition period over a 30.5 kHz spectral width in 16 transients, with a 3 s recycle delay. $^{13}$C MAS NMR spectra were collected with a 2.6 μs π/2 pulse at 100.56 MHz for the $^{13}$C nuclei and a 40 ms acquisition period over a 30.5 kHz spectral width. Proton decoupling was performed with an 86 kHz $^{1}$H decoupling radio frequency and a small phase incremental alteration (SPINAL) decoupling pulse sequence. Cross-polarization and MAS (CP-MAS) NMR studies were conducted with a 5.0 s recycle delay in 1024 transients using a ramped-amplitude pulse sequence with a 2 ms contact time and a 2.9 μs π/2 pulse for the $^{1}$H nuclei. The amplitude of the $^{1}$H nuclei decreased linearly by 92.6% of its final value during the CP contact time. Pulse saturation transfer and MAS (PST-MAS) NMR measurements were carried out in 2048 transients with a 2.6 μs π/2 pulse for the $^{13}$C nuclei with a 5 s recycle delay after the saturation of the $^{1}$H nuclei with 12 consecutive 2.5 μs pulses and a 27.5 μs delay.

### 2.5. Contact Angle Measurements

Pure water (2 μL) was dropped on the wood surface at 25 °C. The contact angles were measured after 1 s and every 10 s thereafter until 1 min. The average value was calculated by measuring at least three different points on the surface of the wood sample.

*2.6. Heat Treatment of the Benzylated Surface*

A small test piece (5 mm × 5 mm × 5 mm) cut from the benzylated wood sample was subject to heat treatment. The test piece was placed on a hot plate at 150 °C with the benzylated surface facing down, a weight of about 3 g was placed on it, and it was heated for 1–10 min. After cooling to 25 °C in the drying chamber, it was subjected to various experiments.

*2.7. Scanning Electron Microscopy (SEM)*

Wood surfaces were observed using scanning electron microscopy (SEM, JSM-IT500; JEOL Ltd., Tokyo, Japan) with an Au coating over the surface (Quick Coater VPS-020; ULVAC Ltd., Yokohama, Japan).

### 3. Results and Discussion

*3.1. Benzylation of Wood*

The wood surface was treated under various conditions. Table 1 summarizes sample codes and the reaction conditions, including the species of alkaline aqueous solutions and benzylating reagents, reaction temperature, and reaction time. Alkaline pretreatments were conducted at 25 °C regardless of the alkaline species.

Figure 2 shows the appearance of a part of the wood samples before and after the benzylation treatments (the photographs of the other samples are shown in Supplementary Figure S1). Entry 1, which was produced using the conventional method, was whitened using benzylation treatment. For most of the other cases, the benzylation reaction did not significantly change the color of the wood samples. This means that avoiding prolonged heating and stirring tends to preserve the original appearance of the wood. Only in case of entry 9, which was prepared using a relatively long alkaline treatment, slight yellowness of the sample is observed (Supplementary Figure S1).

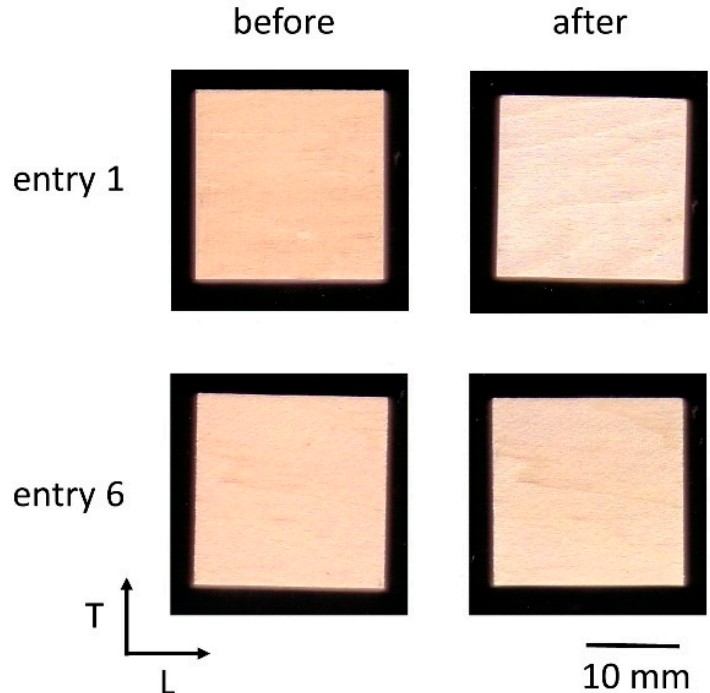

**Figure 2.** Photographs of block-shaped woods before and after benzylation with conventional (entry 1) or novel (entry 6) method.

### 3.2. Chemical Analysis of Benzylated Woods

Figure 3 shows the ATR-IR spectra of the precursor (degreased wood, entry 0) and benzylated wood samples. The intensity of these spectra was standardized based on the peak intensity derived from the C-O single bond appearing at 1060 cm$^{-1}$. The peak intensity of the hydroxyl groups (3100–3600 cm$^{-1}$) decreased significantly after the benzylation treatment. However, IR peaks attributed to the benzyl groups appeared at 736 and 695 cm$^{-1}$ [15]. In addition, IR peaks between 3090 and 3063 cm$^{-1}$, related to the aromatic C-H deformation [16], were observed in some wood samples (Figure 3, entries 1 and 6–9). These changes indicate that the hydroxyl groups were replaced with benzyl groups. According to our previous study, the relative peak intensities in the ATR-IR spectra accurately reflect the reactivity of benzylation [13]. Therefore, the decrease in the rate of the hydroxyl group-derived peaks and the increase in the rate of the aromatic ring-derived peaks were calculated. The decrease rate of the OH-derived peaks was calculated based on the ratio of the peak intensity obtained from the OH stretching vibrations at 3000–3600 cm$^{-1}$ to the peak intensity obtained from the CH stretching vibrations at 2800–3000 cm$^{-1}$ (OH/CH intensity ratio). The increase rate of the benzene ring-derived peaks was calculated based on the ratio of the peak intensity obtained from the mono-substituted benzene rings at 680–714 cm$^{-1}$ to the peak intensity obtained from the CH stretching vibrations (Bn/CH intensity ratio). The lower the OH/CH intensity ratio and the higher the Bn/CH intensity ratio, the higher the number of hydroxyl groups replaced with benzyl groups. These results are presented in the right column of Table 1.

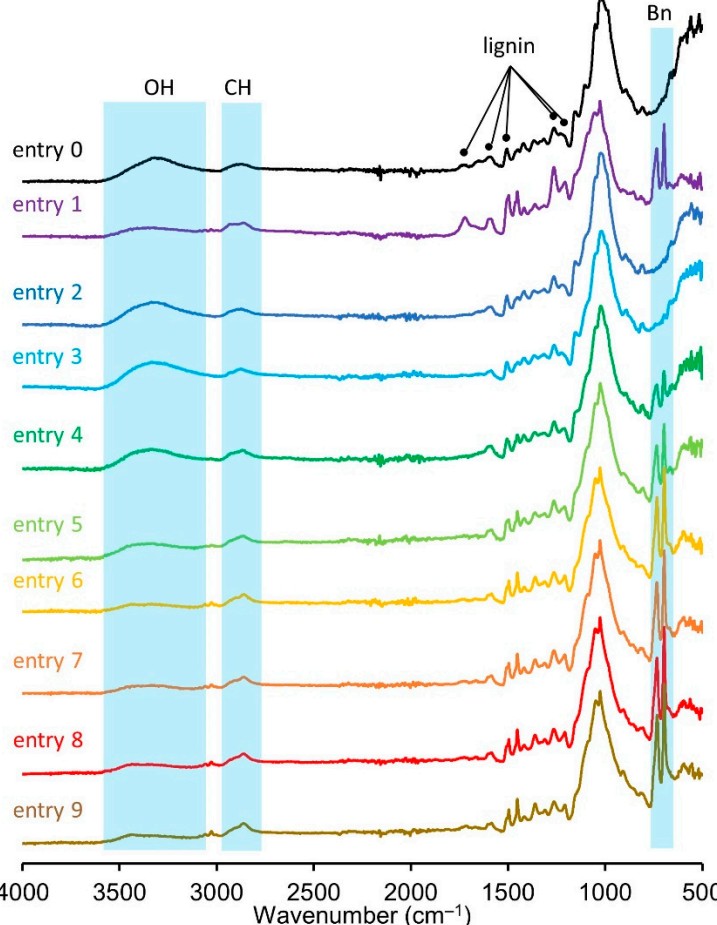

**Figure 3.** ATR-IR spectra of degreased wood (entry 0, black) and benzylated woods (entry 1, purple; entry 2, blue; entry 3, light blue; entry 4, green; entry 5, light green; entry 6, yellow; entry 7, orange; entry 8, red; and entry 9, brown).

The OH/CH ratio and the Bn/CH ratio of the degreased wood, i.e., entry 0, were 3.6 and 0.3, respectively. Entry 1, prepared via the conventional method with heating and stirring, was significantly benzylated, and the OH/CH ratio and Bn/CH ratio were 0.7 and 9.4, respectively. However, for entry 1, the intensity of the IR peaks derived from lignin, which appeared at 1720, 1600–1585, 1500, and 1260–1205 $cm^{-1}$, was higher than that for the other samples. This suggests that some carbohydrates were decomposed by heating the wood for a long time after immersing it in a strong alkaline aqueous solution. This carbohydrate chain decomposition might cause a relative increase in the peak intensity derived from lignin. The results of the solid-state NMR analysis also supported the abovementioned hypothesis, as described later. Next, we attempted the benzylation of wood with the same reagents as for entry 1 at 25 °C (entry 2). However, the values of the OH/CH ratio and the Bn/CH ratio were similar to those for entry 0, implying negligible progress in benzylation. A similar result was obtained for entry 3 using BnBr as the benzylation reagent. However, for entry 4, the OH/CH ratio decreased to 1.7 and the Bn/CH ratio increased to 3.1, although all the experimental conditions, except for the pretreatment, remained the same as those for entry 2. This suggests that the use of 50% [($n$-Bu)$_4$P]OH promotes benzylation to a certain extent, even without heating. When the reaction time was set as 30 min, benzylation progressed further (entry 5). Moreover, for entry 6, where BnBr was used as the benzylating reagent, the OH/CH ratio was 0.6 and the Bn/CH ratio reached 8.1, indicating that benzylation proceeded efficiently. For entry 3, benzylation progressed negligibly, despite the use of BnBr. Therefore, the pretreatment with 50% [($n$-Bu)$_4$P]OH was deemed as the main reason for the efficient benzylation of wood without heating. Furthermore, according to the results of $t$-test, there was negligible change in the degree of benzylation when the treatment time was varied ($p = 0.80$ and $0.28$ for Bn/CH values between entry 6 and 7, 6 and 8, respectively). This indicates that the benzylation reaction proceeded under a very short non-heat treatment.

Solid-state NMR spectroscopy can provide information regarding the chemical changes in each substituent of the constituent polymers (cellulose, hemicellulose, and lignin) in woody materials. Here, entry 1, prepared via the conventional method, and entry 6, prepared via the new method, were subjected to solid-state NMR spectroscopy measurements. As a reference, the same measurements were conducted for entry 0, which had only been degreased.

As demonstrated in our previous study on the chemical modification of Japanese cypress [17], hydrophobicity and moisture balance can be monitored using the $^1$H MAS NMR method. Figure 4 shows the $^1$H MAS NMR spectra of the degreased and benzylated wood plates in the air-dried state. The degreased wood plate (entry 0) exhibited a large broad-singlet peak around 0–10 ppm, which was attributed to the overlapping of constituent polymers and the moisture in the wood plate. After benzylation using both the conventional (entry 1) and new (entry 6) methods, this large signal derived from water molecules decreased, and the benzyl signal newly appeared in the aromatic region (at approximately 7 ppm). Based on the $^1$H MAS NMR spectra of modified wood in the heat-dried state [17], benzylation improved the hydrophobicity of the wood plate, as evidenced by the lack of water molecules.

The changes in the constituent polymers of the wood plate can be monitored using the $^{13}$C CP-MAS NMR spectra, which provide more detailed information regarding each substituent, as compared with the $^1$H MAS NMR spectra [17]. Figure 5 shows the $^{13}$C CP-MAS NMR spectra of the degreased and benzylated wood plates. After benzylation, the ring signals (1-C$_6$H$_5$: 134 ppm and 2,3,4,5,6-C$_6$H$_5$: 128 ppm) in the benzyl group newly appeared when using either the conventional method (entry 1) or the new method (entry 6). Simultaneously, the signals of carbohydrates were reduced by benzylation, especially the C1 and C6(C5) signals, owing to the degradation of carbohydrates. Although the ATR-IR spectra showed an increase in the lignin G ring and C=O peaks when using the conventional method (entry 1), the increase in the signals of these groups was more gradual in the $^{13}$C CP-MAS NMR spectra. This is because the ATR-IR spectra can detect the

changes in the very shallow regions of the surface, whereas solid-state NMR spectroscopy necessitates a sample with a certain thickness of the surface of the wood plate. The results pertaining to the depth of benzylation are described later.

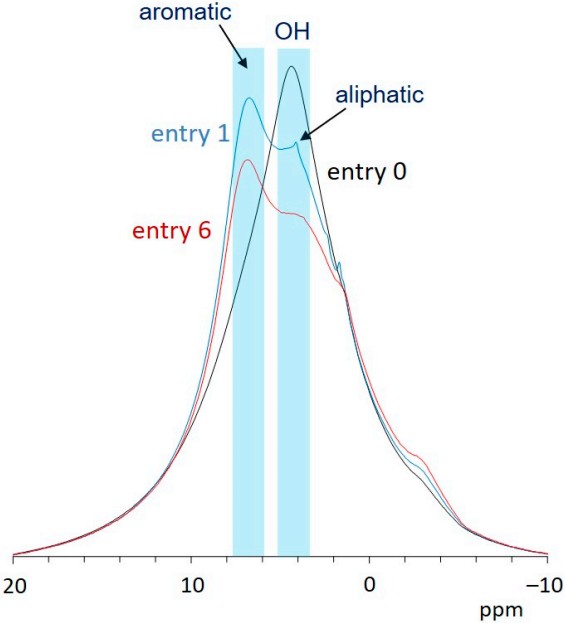

**Figure 4.** $^{1}$H MAS NMR spectra of degreased wood (entry 0, black) and benzylated woods (entry 1, blue; and entry 6, red).

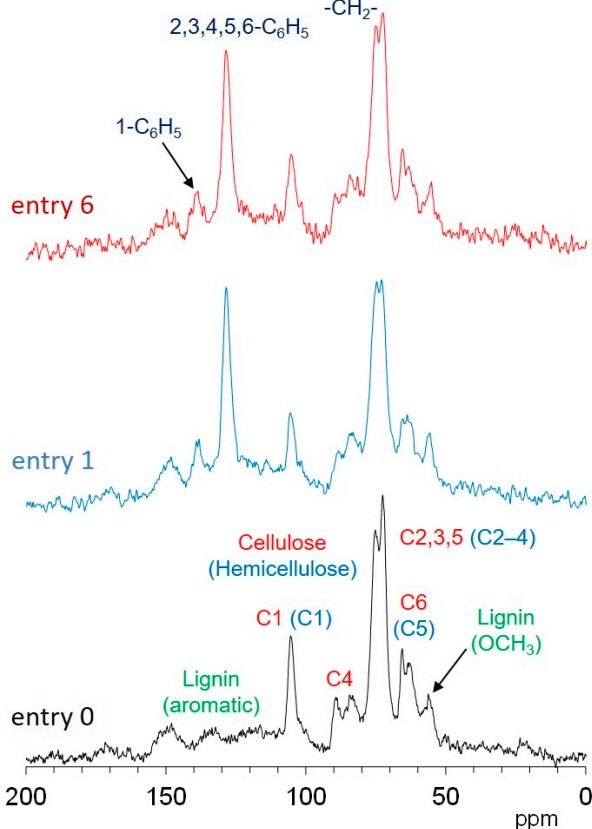

**Figure 5.** $^{13}$C CP-MAS NMR spectra of degreased wood (entry 0, black) and benzylated woods (entry 1, blue; and entry 6, red).

The $^{13}$C PST-MAS NMR method enhances the signals of the more flexible portions with high molecular mobilities owing to the nuclear Overhauser effect for carbon atoms connected with hydrogen atoms. Therefore, a comparison of the signal intensities between the $^{13}$C CP-MAS and $^{13}$C PST-MAS NMR spectra can provide information regarding the molecular mobility of each substituent [18]. Figure 6 shows the $^{13}$C PST-MAS NMR spectra of the degreased and benzylated woods. As shown in the $^{13}$C CP-MAS NMR spectra of benzylated wood, the signal intensity of 2,3,4,5,6-$C_6H_5$ is almost the same for both the conventional method and the new method. Nevertheless, as indicated by the $^{13}$C PST-MAS NMR spectra of benzylated wood, the 2,3,4,5,6-$C_6H_5$ signal when using the conventional method (entry 1) is clearly larger than that when using the new method (entry 6). This indicates that the molecular mobility of the benzyl group increased to a greater extent when using the conventional method, as compared with that when using the new method. In other words, benzylation using the conventional method (entry 1) is detrimental to carbohydrates owing to the severe reaction conditions (long heating time with strong alkaline). By contrast, benzylation using the new method (entry 6) was milder and prevented the degradation of the carbohydrates. The $^{13}$C PST-MAS NMR results were consistent with the ATR-IR results indicating the G ring stretching of lignin, considering the relative reduction in the carbohydrate peaks.

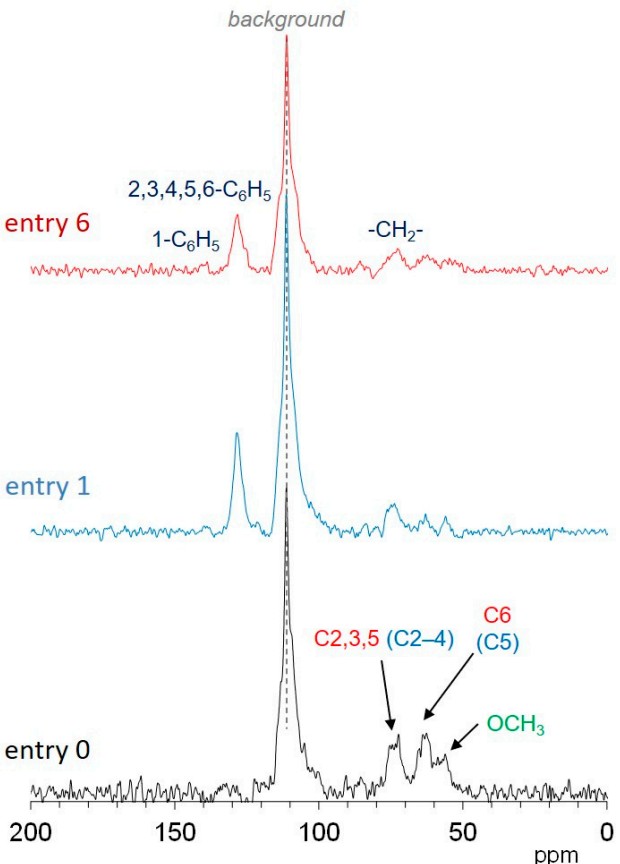

**Figure 6.** $^{13}$C PST-MAS NMR spectra of degreased wood (entry 0, black) and benzylated woods (entry 1, blue; and entry 6, red).

Based on the results of these chemical analyses, the novel benzylation process involving pretreatment with [($n$-Bu)$_4$P]OH and non-heated benzylation treatment with BnBr was concluded to be an excellent method for the benzylation of wood surfaces, without significant decomposition of the wood components. Therefore, in the subsequent evaluations of physical properties, we focused on entry 6 prepared using the new method.

### 3.3. Hydrophobicity of Benzylated Wood

To evaluate hydrophobicity, the water contact angles of the benzylated surface were measured (Figure 7). For selected cases, images of the contact angle measurements are presented in Figure 8. The contact angle for entry 0, where only degreasing treatment was performed, was 87°, 1 s after dropping (Figure 7, black). The water droplets quickly permeated the wood, and the water was completely absorbed after 50 s. By contrast, for entry 6, which had a high degree of benzylation, the water contact angle after dropping exceeds 110°, and this value was greater than 65° even after 1 min (Figure 7, orange). This result confirmed that wood can be imparted with hydrophobicity via the non-heating benzylation treatment.

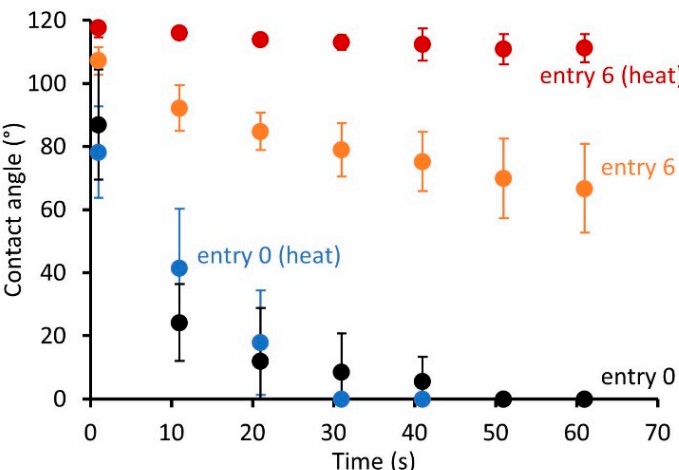

**Figure 7.** Water contact angles on the degreased wood (entry 0) and benzylated wood (entry 6) surfaces before and after heat treatment. Error bar shows the standard deviations based on the values of 3–5 samples.

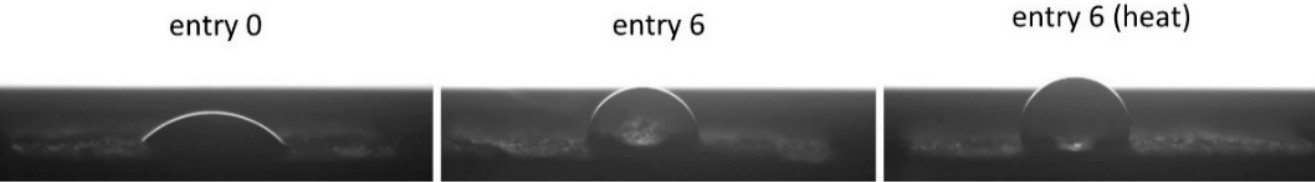

**Figure 8.** Contact angles for water droplets after 11 s on degreased wood (entry 0), benzylated wood (entry 6), and heat-treated benzylated wood (entry 6 (heat)) surfaces.

Next, the surface of the benzylated wood was heat treated for further improving its hydrophobicity. Highly benzylated wood is known to exhibits thermoplasticity [13,15,19]. It has also been reported that, on benzylating lignocellulose fiber and then heat treating it, the benzylated fiber melted and became flat [20]. If the surface of the benzylated block-shaped wood prepared in this study can be changed smoothly via heat treatment, the absorption of water droplets into the wood will be hindered and greater water resistance may be imparted to the wood surface. Therefore, the thermoplasticity of entry 6 was evaluated using dynamic mechanical analysis. As a result, a softening point was identified near 100 °C, thus confirming that it possesses thermoplasticity (Supplementary Figure S2). Therefore, the benzylated surface of entry 6 was subjected to heat treatment. To ensure that the treatment of the wood surface was as simple and quick as possible, the heat treatment of entry 6 was conducted within a few minutes. During the heat treatment, a significantly small load was applied to ensure appropriate contact between the benzylated surface and the hot plate. The values of the water contact angle for the obtained wood sample and photographs of the water droplet during the measurements are presented in

Figures 7 and 8 (entry 6 (heat)). After the heat treatment, the contact angle of the water droplet for entry 6 reached approximately 120° (Figure 7, red). Furthermore, this value remained essentially unchanged even after 1 min. An additional sample was prepared and subjected to heat treatment for 10 min. However, the water contact angle remained the same (Supplementary Figure S3, $p > 0.48$ at any points on the $t$-test). This result suggests that the hydrophobization of the benzylated wood surface via heat treatment was completed within 5 min. In a case of poplar wood, it has been reported that heat treatment at 160–220 °C for several hours increased the hydrophobicity of wood [21,22]. As well as these previous reports, the non-benzylated cedar (entry 0) was also heat-treated at 150 °C for 5 min, and the water contact angle was investigated (Figure 7, blue). According to the result of $t$-test, the water contact angle of entry 0 did not change before and after the heat treatment ($p = 0.48$ at 1 s, and $p = 0.15$ at 11 s). This result showed that the hydrophobicity of non-benzylated wood did not change even after the heat treatment at 150 °C in this experiment. Therefore, the increase in the water contact angle of entry 6 with heat-treatment was derived from the benzyl groups introduced into the wood surface.

Based on the above-mentioned results, it was found that block-shaped wood with an excellent water-resistant surface can be obtained using the two rapid treatments: easy benzylation and subsequent heat treatment. In addition, no coloring or cracking of the wood occurred before and after the heat treatment, and there was no change in appearance (Supplementary Figure S4). Furthermore, on lightly tracing the surface of the wood after heat treatment with a finger, there were no changes in the feel. The ATR-IR spectra of entry 6 before and after heat treatment were also almost the same (Supplementary Figure S5).

To elucidate the depth up to which wood should be benzylated in order to achieve the above-mentioned excellent hydrophobicity, the depth of benzylation for the wood surface in entry 6 was evaluated. As shown in Figure 9, the surface of entry 6 was cut diagonally, and ATR-IR spectra were measured at various points. By calculating the depth from the wood surface ($d$) at each measured point and examining the relationship with the value of the Bn/CH ratio, the depth up to which benzylation had progressed was determined. Based on these examinations, it was found that only the areas in close vicinity to the surface were benzylated (Figure 10). The Bn/CH ratio at a depth of approximately 50 μm from the benzylated surface was less than half that at the surface. In addition, the Bn/CH ratio at approximately 200 μm from the surface was almost the same as that of degreased wood. This implies that only approximately 1–2 cells near the surface were benzylated and the internal cell walls at greater depths underwent negligible benzylation. The above results indicate that a significantly low thickness of the benzylated layer from the surface is required to obtain hydrophobic wood with a water contact angle exceeding 110°. This suggests that the wood surface can be hydrophobized by simply applying a chemical solution, washing, and a short heat treatment, given that benzylation up to greater depths is not required. This may facilitate the hydrophobization of wood with a large surface area. This is difficult to achieve using the conventional method, which requires prolonged periods of heating and stirring.

To elucidate why the benzylated layer near the surface alone led to an improvement in the hydrophobicity of entry 6 under heat treatment, SEM was used to evaluate the surface of the benzylated wood before and after heat treatment (Figure 11). Thus, it was confirmed that, after heat treatment, the surface of entry 6 featured less unevenness than that before heat treatment. The reproducibility was confirmed by performing similar SEM observations on various locations in the several samples prepared under the same conditions. The results are shown in Supplementary Figure S6. Owing to the cutting of wood, many cells on the surface of the wood form U-shaped walls, as shown in Figure 11. In the enlarged SEM image, innumerable U-shaped cell walls can be observed on the wood surface before the heat treatment. However, these U-shaped cell walls are changed to a flatter shape after heat treatment. Owing to this reduction in the unevenness of the wood surface, water absorption is suppressed, and a water contact angle of 110° or higher can be maintained. Based on these results, for entry 6, which underwent simple heat treatment,

the wood surface was found to be smoothed at a micro level that cannot be detected by humans via visual inspection or touch. Nevertheless, this smoothing effect improved the hydrophobicity of the benzylated wood.

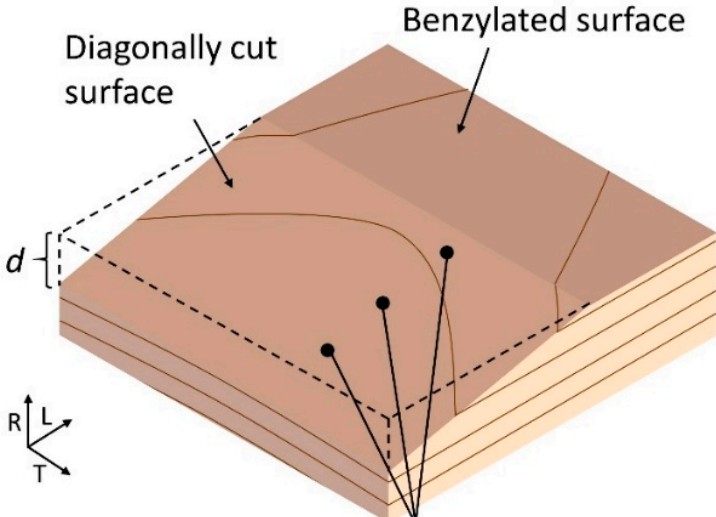

**Figure 9.** Schematic of cutting the wood diagonally and measuring the ATR at various points.

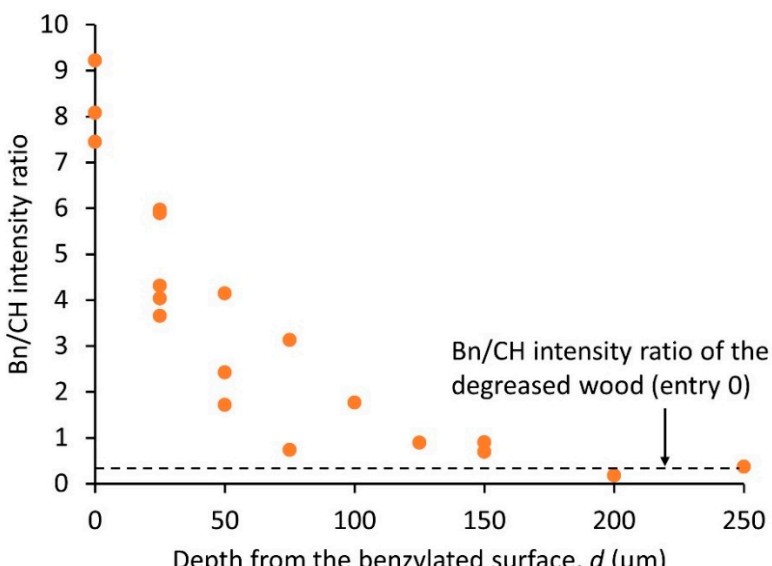

**Figure 10.** Relationship between the distance from the benzylated surface of entry 6 and the Bn/CH ratio at that point.

The newly developed wood hydrophobization technique can be implemented by a very easy and rapid process. Using this technique, significant hydrophobization was achieved by treating only the very vicinity of the wood surface. Therefore, the amount of reagent required is very small as compared with conventional methods. It also has the advantage that [(*n*-Bu)$_4$P]OH can be recovered and reused. Furthermore, since the benzylated area of the wood is limited to the surface only, most of the area can be recycled or discarded as untreated wood. In summary, the wood hydrophobization technique developed in this study is an environmentally friendly chemical treatment process. In the near future, the properties of the benzylated block-shaped wood will be further analyzed such as the weather resistance, mechanical strength, dimensional stability, and biodeterioration resistance, e.g., ant and mold resistances. In addition, we plan to investigate the

effectiveness of this hydrophobizing process on other lignocellulosic materials, including herbaceous plants such as bamboo, which are expected to be used industrially [23–25].

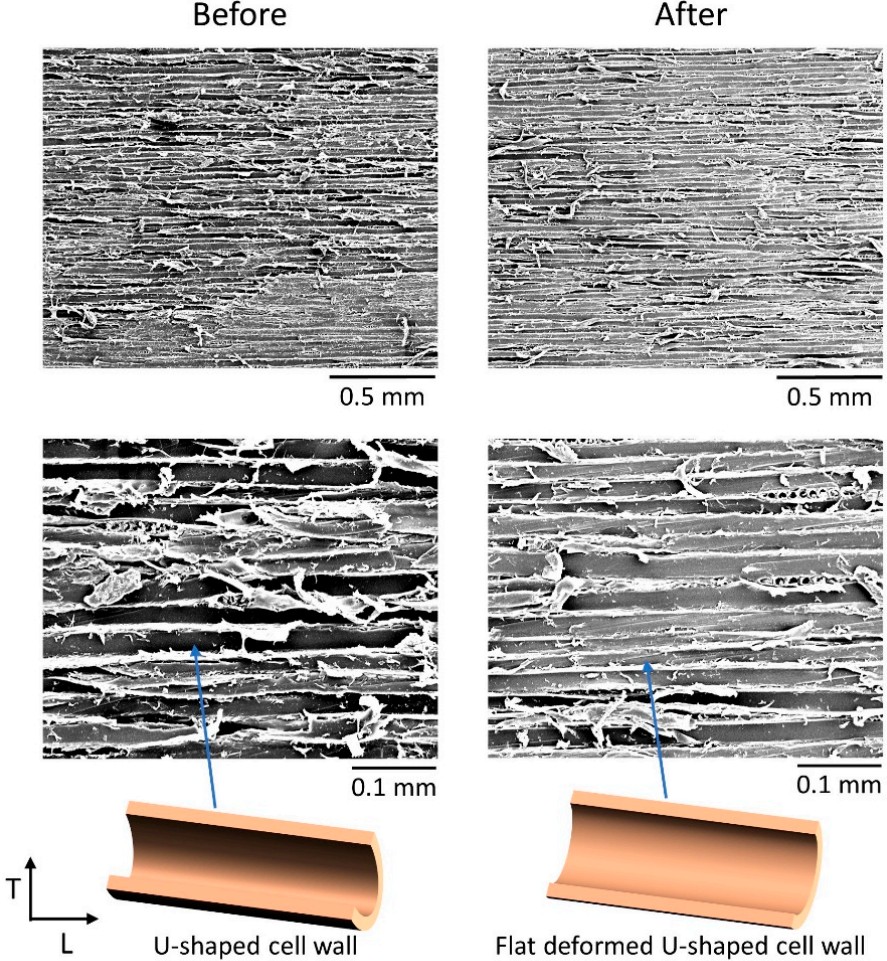

**Figure 11.** SEM images of entry 6 before and after heat treatment, and images of the cell walls on wood surfaces.

## 4. Conclusions

A novel and easy method for the benzylation of block-shaped wood surfaces without heating was developed in this study. Using an aqueous solution of [(*n*-Bu)$_4$P]OH as the pretreatment solvent, benzylation comparable to or better than that afforded by the conventional method could be realized via non-heat treatment for a total of 20 min. The chemical structure of the obtained benzylated wood was analyzed using the ATR-IR and solid-state NMR methods, and it was clarified that the damage to the carbohydrates was less than that in the conventional method. After benzylation, the water resistance of the wood was improved, and the wood also exhibited thermoplasticity. A simple heat treatment for approximately 5 min further improved the hydrophobicity of the benzylated wood, resulting in a water contact angle of 110° or higher, which remained essentially unchanged even after 1 min. A considerably low thickness of the benzylated layer from the surface was required to obtain such hydrophobic wood. The hydrophobicity of the benzylated wood improved under simple heat treatment because the cell walls on the surface of the thermoplastic benzylated wood were altered to a flatter shape. As this novel benzylation method does not require special equipment or long-term heating and stirring, it is expected to be suitable for use as a hydrophobic treatment for wood with large surface areas or for wood products after molding.

**Supplementary Materials:** The following are available online at https://www.mdpi.com/article/10.3390/f12081028/s1, Experimental information of DMA analysis, Figure S1: Photographs of block-shaped wood before and after benzylation, Figure S2: Relative $E'$ of degreased wood (entry 0) and benzylated wood (entry 6) based on $E'$ at 40 °C, Figure S3: Water contact angle of heat-treated entry 6 surface for 0 (orange), 1 (green), 5 (red), and 10 min (purple), Figure S4: Photographs of a piece of entry 6 before and after heat treatment, Figure S5: ATR-IR spectra of entry 6 before and after heat treatment, Figure S6: SEM images on various locations in the several samples prepared under the same conditions as entry 6 before and after heat treatment.

**Author Contributions:** Conceptualization, M.A.; methodology, M.A., M.S., T.M. and M.N.; formal analysis, M.A., M.S., T.M. and M.N.; investigation, M.A.; data curation, M.A. and M.N.; writing—original draft preparation, M.A.; writing—review and editing, M.S., T.M. and M.N.; project administration, M.A.; funding acquisition, M.A. and T.M. All authors have read and agreed to the published version of the manuscript.

**Funding:** This work was financed by Grants-in-Aid (KAKENHI) for Young Scientist Research (No. 19K15890 to MA) and for the Promotion of Joint International Research (Fostering Joint International Research (A) 18KK0417 to TM) from the Japan Society for the Promotion of Science (JSPS).

**Data Availability Statement:** The data presented in this study are available upon request from the corresponding author.

**Acknowledgments:** The authors wish to thank the Japan Society for the Promotion of Science (JSPS) for their financial support of this work.

**Conflicts of Interest:** The authors declare no conflict of interest.

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
