# Peer review of "Surface Hydrophobization of Block-Shaped Wood with Rapid Benzylation"

_forests, doi:10.3390/f12081028_

Round 1
Reviewer 1 Report
I have found this research very decent and informative. However, I believe the manuscript could have been prepared in a more declarative way, which boosts the significance of this study.
Here are two major comments:
I. recommend that authors particularly justify why they used this chemical/process ( benzylation) and compare its pros and cons with other chemicals/processes.
2. I recommend that the authors indicate the retention of chemical preservatives in the treated specimens.
Here are three minor comments:
- Authors are encouraged to provide more information about the selected wood species for this study, especially in terms of location (region). Also, the authors can add more information about the particular application of Japanese Cedar (domestic and overseas).
- Authors are encouraged to compare the environmentally-friendly features of their material/process to those of previous studies. References could be included but are not limited to: Influence of copper and biopolymer/Saqez resin on the properties of poplar wood H Dong, M Bahmani, S Rahimi, M Humar Forests 11 (6), 667.
- Authors are encouraged to compare the wettability of the treated samples to that of previous studies. References could be included but are not limited to: Effect of heat treatment on bonding performance of poplar via insight into dynamic wettability and surface strength transition from outer to inner layers D Chu, J Mu, S Avramidis, S Rahimi, Z Lai, S Ayanleye Holzforschung 74 (8), 777-787.
Author Response
Dear Reviewer,
Thank you for your appreciation.
I have found this research very decent and informative. However, I believe the manuscript could have been prepared in a more declarative way, which boosts the significance of this study.
Ans. Thank you for your positive comments. We revised our manuscript according to your helpful comments.
Here are two major comments:
- I recommend that authors particularly justify why they used this chemical/process (benzylation) and compare its pros and cons with other chemicals/processes.
Ans. We added sentences about the pros and cons of benzylation compared to the acetylation of wood in Introduction as follows:
“Among various chemical modifications, the acetylated wood has been studies most actively and is being put into practical uses. Compared to the acetylated wood, the benzylated wood has advantages such as high hydrophobicity and ultraviolet absorption capacity. However, the benzylation is less reactive than the acetylation and requires a longer stirring at higher temperature.” (Line 83–87 in the revised manuscript)
- I recommend that the authors indicate the retention of chemical preservatives in the treated specimens.
Ans. Benzylated wood is known to have good biodeterioration resistance. It is one of the reasons that we chose the benzylation to wood among various chemical modifications (please see Line 81–83 in the revised manuscript). Near future, we will examine the biodeterioration resistance of the surface benzylated wood produced by the process developed in this study. We added a sentence related to this topic in the revised manuscript as follows:
“In the near future, the properties of the benzylated block-shaped wood will be further analyzed such as the weather resistance, mechanical strength, dimensional stability, and biodeterioration resistance, e.g. ant and mold resistances.” (Line 424–426 in the revised manuscript)
Here are three minor comments:
- Authors are encouraged to provide more information about the selected wood species for this study, especially in terms of location (region). Also, the authors can add more information about the particular application of Japanese Cedar (domestic and overseas).
Ans. We added the information about the location and the particular application of Japanese cedar in Materials and Methods as follows:
“As a block-shaped wood, Japanese cedar (Cryptomeria japonica) was used because it is the most common coniferous wood as a structural material in Japan among various types of wood. The original Japanese cedar timber was logged in Kyoto prefecture and was purchased from Science Shokai, LLC. (Nagoya, Japan).” (Line 114–117 in the revised manuscript)
- Authors are encouraged to compare the environmentally-friendly features of their material/process to those of previous studies. References could be included but are not limited to: Influence of copper and biopolymer/Saqez resin on the properties of poplar wood H Dong, M Bahmani, S Rahimi, M Humar Forests 11 (6), 667.
Ans. We added some references as examples of the environmentally friendly hydrophobization of wood in Introduction. (Line 41,42 in the revised manuscript)
In addition, we added several sentences in the last of Results and Discussions as follows:
“The newly developed wood hydrophobization technique can be implemented by a very easy and rapid process. Using this technique, significant hydrophobization was achieved by treating only the very vicinity of the wood surface. Therefore, the amount of reagent required is very small as compared with conventional methods. It also has the advantage that [(n-Bu)4P]OH can be recovered and reused. Furthermore, since the benzylated area of the wood is limited to the surface only, most of the area can be recycled or discarded as untreated wood. In summary, the wood hydrophobization technique developed in this study is an environmentally friendly chemical treatment process.” (Line 417–424 in the revised manuscript)
- Authors are encouraged to compare the wettability of the treated samples to that of previous studies. References could be included but are not limited to: Effect of heat treatment on bonding performance of poplar via insight into dynamic wettability and surface strength transition from outer to inner layers D Chu, J Mu, S Avramidis, S Rahimi, Z Lai, S Ayanleye Holzforschung 74 (8), 777-787.
Ans. We added several sentences and two references about the hydrophobization effect of heat treatment of natural wood. In addition, the water contact angle of entry 0, non-benzylated wood, after heat treatment was also measured additionally. According to this result, the water contact angle of entry 0 was not changed even after heating at 150 °C for 5 min. These results and discussion were added to the revised manuscript as follows:
“In a case of poplar wood, it has been reported that heat treatment at 160–220 °C for several hours increased the hydrophobicity of wood [21,22]. As well as these previous reports, the non-benzylated cedar (entry 0) was also heat-treated at 150 °C for 5 min, and the water contact angle was investigated (Figure 7, blue). According to the result of t-test, the water contact angle of entry 0 did not change before and after the heat treatment (p = 0.48 at 1s, and p = 0.15 at 11 s). This result showed that the hydrophobicity of non-benzylated wood did not change even after the heat treatment at 150 °C in this experiment. Therefore, the increase in the water contact angle of entry 6 with heat-treatment was derived from the benzyl groups introduced into the wood surface.” (Line 357–365 in the revised manuscript)

Reviewer 2 Report
The work presented here addresses a current topic related to improving the performance of wood. In the introduction chemical modifications are presented, from my point of view more general information about modifications (chemical, thermal) and principles of their operation are missing. Have the authors considered how to dispose of the waste produced during the manufacture of products from such modified wood and of the products themselves after their useful life? From the point of view of green chemistry and responsible economy, it would be good to propose future possibilities of their management. And the possible impact on the environment.
The results presented lack information about the standard deviation of the results obtained. Such information would improve the quality of the article.
In order to improve the quality of the publication, I would suggest performing a significance of differences test for the results obtained. This may help to indicate the direction of research in the future and will significantly contribute to the quality of publications.
Author Response
Dear Reviewer,
Thank you for your appreciation.
The work presented here addresses a current topic related to improving the performance of wood. In the introduction chemical modifications are presented, from my point of view more general information about modifications (chemical, thermal) and principles of their operation are missing.
Ans. General information about chemical and thermal modification of wood including the principles of their operation were added to the Introduction section and some references were cited as follows:
“To overcome these problems, various modifications have been studied and some results have been achieved [1–3]. However, many issues still remain, such as the complexity of processing and the environment impact of the process.” (Line 32–34 in the revised manuscript)
Have the authors considered how to dispose of the waste produced during the manufacture of products from such modified wood and of the products themselves after their useful life? From the point of view of green chemistry and responsible economy, it would be good to propose future possibilities of their management. And the possible impact on the environment.
Ans. Thank you for your very important advice. We added several sentences in the last of Results and Discussions as follows:
“The newly developed wood hydrophobization technique can be implemented by a very easy and rapid process. Using this technique, significant hydrophobization was achieved by treating only the very vicinity of the wood surface. Therefore, the amount of reagent required is very small as compared with conventional methods. It also has the advantage that [(n-Bu)4P]OH can be recovered and reused. Furthermore, since the benzylated area of the wood is limited to the surface only, most of the area can be recycled or discarded as untreated wood. In summary, the wood hydrophobization technique developed in this study is an environmentally friendly chemical treatment process.” (Line 416–423 in the revised manuscript)
The results presented lack information about the standard deviation of the results obtained. Such information would improve the quality of the article.
Ans. Because wood is a natural resource and not a perfectly homogeneous material, all researches include some variation in their results. Therefore, as you pointed out, the standard deviation information is very important. We calculated the standard deviation for the results obtained in this study. The obtained information was summarized in Table 1 and Figure 6.
In order to improve the quality of the publication, I would suggest performing a significance of differences test for the results obtained. This may help to indicate the direction of research in the future and will significantly contribute to the quality of publications.
Ans. Thank you for your helpful advice. Following your advice, t-test was conducted on some data in this study. In addition, we revised several sentences in Results and Discussion as follows:
“Furthermore, according to the results of t-test, there was negligible change in the degree of benzylation when the treatment time was varied (p = 0.80 and 0.28 for Bn/CH values between entry 6 and 7, 6 and 8, respectively); this indicates that the benzylation reaction proceeded under a very short non-heat treatment.” (Line 248–251 in the revised manuscript)
“An additional sample was prepared and subjected to heat treatment for 10 min; however, the water contact angle remained the same (Figure S3 in SI, p > 0.48 at any points on the t-test)” (Line 354–356 in the revised manuscript)
“According to the result of t-test, the water contact angle of entry 0 did not change before and after the heat treatment (p = 0.48 at 1s, and p = 0.15 at 11 s).” (Line 361–363 in the revised manuscript)

Reviewer 3 Report
This manuscript describes a method for surface hydrophobization of block-shaped wood with rapid benzylation. The results are well organized and can support the conclusions very well. Major revision is requested before its publication. My comments are as below:
- This work uses tetra-n-butylphosphonium hydroxide as a pretreatment solvent. However, It causes corrosive effects on skin and mucous membrane and irritates skin and mucous membrane. Then, how can the productors or users be present from this dangerous reagent?
- The modification diagrammatic sketch should be given for the benzylation processes.
- For Figure 10. I think that the differences between the U-shaped cell wall as the authors state are just depend on how much they cut off along the radial direction. Because the SEM images are for different samples. Only the differences between the same areas before and after the treatment are convincing.
- As wood, bamboo-based products also need hydrophobicity in some specific application fields. Can the hydrophobization treatments reported in this manuscript also be suitable for bamboo? The authors are suggested to give some discussions on this based on some references about bamboo processing. For example, Journal of Renewable Materials, 2021, 9(5):959-977. DOI: 10.32604/jrm.2021.014285; European Journal of Wood and Wood Products, 2021, DOI: 10.1007/s00107-021-01722-1.
Author Response
Dear Reviewer,
Thank you for your appreciation.
This manuscript describes a method for surface hydrophobization of block-shaped wood with rapid benzylation. The results are well organized and can support the conclusions very well. Major revision is requested before its publication. My comments are as below:
Ans. Thank you for your positive comments. We revised our manuscript according to your comments.
This work uses tetra-n-butylphosphonium hydroxide as a pretreatment solvent. However, It causes corrosive effects on skin and mucous membrane and irritates skin and mucous membrane. Then, how can the productors or users be present from this dangerous reagent?
Ans. It is relatively safe because it can be used in smaller amounts compared to NaOH and the risk of contact is reduced by optimizing the process. We added some relevant sentences to Results and Discussion in the revised manuscript as follows:
“The newly developed wood hydrophobization technique can be implemented by a very easy and rapid process. Using this technique, significant hydrophobization was achieved by treating only the very vicinity of the wood surface. Therefore, the amount of reagent required is very small as compared with conventional methods. It also has the advantage that [(n-Bu)4P]OH can be recovered and reused. Furthermore, since the benzylated area of the wood is limited to the surface only, most of the area can be recycled or discarded as untreated wood. In summary, the wood hydrophobization technique developed in this study is an environmentally friendly chemical treatment process.” (Line 417–424 in the revised manuscript)
The modification diagrammatic sketch should be given for the benzylation processes.
Ans. The modification diagrammatic sketch of the benzylation processes was added to the revised manuscript. (Figure 1, Line 137,138 in the revised manuscript)
For Figure 10. I think that the differences between the U-shaped cell wall as the authors state are just depend on how much they cut off along the radial direction. Because the SEM images are for different samples. Only the differences between the same areas before and after the treatment are convincing.
Ans. The wood surface has to be coated by Au for SEM observation. Therefore, we cannot perform SEM observation before and after heat treatment at the same location of the exact same sample. Therefore, we confirmed the reproducibility by performing similar SEM observations on various locations in the several samples prepared under the same conditions. The results are shown in Supplementary Information (Figure S6). As a result, it was found that the flat U-shaped cell wall was contained more abundantly after the heat treatment than before the heat treatment at any point of any lot. From the above results, it is probable that the reproducibility could be confirmed. We added several sentences about the confirmation of the reproducibility to the revised manuscript as follows:
“The reproducibility was confirmed by performing similar SEM observations on various locations in the several samples prepared under the same conditions. The results are shown in Figure S6 in SI.” (Line 402–404 in the revised manuscript)
As wood, bamboo-based products also need hydrophobicity in some specific application fields. Can the hydrophobization treatments reported in this manuscript also be suitable for bamboo? The authors are suggested to give some discussions on this based on some references about bamboo processing. For example, Journal of Renewable Materials, 2021, 9(5):959-977. DOI: 10.32604/jrm.2021.014285; European Journal of Wood and Wood Products, 2021, DOI: 10.1007/s00107-021-01722-1.
Ans. Bamboo is also one of the renewable resources that is expected to be effectively used as an industrial material. We think that the hydrophobization method developed in this paper is applicable to bamboo, too. We added a relavant sentence in Results and Discussion to the revised manuscript as follows:
“In addition, we plan to investigate the effectiveness of this hydrophobizing process on other lignocellulosic materials including herbaceous plants such as bamboo, which are expected to be used industrially [23–25].” (Line 427–429 in the revised manuscript)

Round 2
Reviewer 1 Report
The authors addressed my comments satisfactorily and made the required changes accordingly.
Reviewer 3 Report
Acceptable as it is now.